# A Green Supply-Chain Decision Model for Energy-Saving Products That Accounts for Government Subsidies

**Jian Xue** [1], **Ruifeng Gong** [1,2,*], **Laijun Zhao** [3,4,*], **Xiaoqing Ji** [1] and **Yan Xu** [1]

[1] School of Economics and Management, Shaanxi University of Science and Technology, Xi'an 710021, China; xuejian@sust.edu.cn (J.X.); m18292088518@163.com (X.J.); 15029883270@163.com (Y.X.)
[2] Business School, Hunan University of Humanities, Science and Technology, Loudi, Hunan 417000, China
[3] Sino-US Global Logistics Institute, Shanghai Jiao Tong University,1954 Huashan Rd., Shanghai 200030, China
[4] Antai College of Economics and Management, Shanghai Jiao Tong University, 1954 Huashan Rd., Shanghai 200030, China
* Correspondence: 13107214247@163.com (R.G.); ljzhao70@sjtu.edu.cn (L.Z.)

**Abstract:** Government subsidies are a common policy adopted to promote energy conservation and emission reduction. The decision-making that occurs within the green supply chain for energy-saving products under government subsidies is an area of great academic interest and game theory is becoming a popular tool in such research. In this paper, we examined centralized and decentralized decision-making models for the green supply chain and a coordinated decision-making model for revenue-sharing contracts based on game theory. We studied the effects of government subsidies on retail prices, energy conservation levels, market demand, supply chain profits, and social welfare for energy-saving products. We then compared the effectiveness of the three models using a numerical example. Our results revealed the range of contract parameters for which manufacturer and retailer profits increase. Our results show that government subsidies can significantly improve social welfare and promote the improvement of energy-saving products. Centralized decision-making generates higher profits than decentralized decisions and government subsidies were positively correlated with the level of energy conservation, product prices, and market demand. Revenue sharing contract coordination decisions can coordinate the supply chain and achieve the same effect as centralized decisions.

**Keywords:** government subsidies; energy conservation; green supply chain; social welfare

## 1. Introduction

From the perspective of both theory and practice, sustainable development is a very important issue. The concept and problems of sustainability are realized in many supply chain relationships [1]. With the government and the public increasingly demanding environmental protection, producers within the product supply chain must increasingly pay attention to their environmental performance [2]. To effectively implement the United Nations Framework Convention on Climate Change, China's government submitted its intended nationally determined contributions and described its enhanced actions to combat climate change [3]. This document promises to reduce carbon dioxide emissions per unit of GDP by 60% to 65% by 2030, compared with 2005 levels. To gain a competitive advantage in the market, producers must develop energy-saving products that meet consumer demand and implement green supply chain management [4]. The production of green products not only requires each node in the production process to be green, but also requires the integration of green procurement, design, production, sales, and recycling within all aspects of product management in a green process [5].

Unfortunately, this additional overhead increases the cost of energy-saving products compared with conventional products [6].

Due to a lack of funds, producers find it difficult to fund research and development (R&D) that leads to green innovation to produce more environmentally friendly products. Therefore, it is difficult for producers of these products to obtain a favorable competitive position in the market, resulting in insufficient motivation for enterprises to produce energy-saving products. To mitigate this problem, governments often utilize subsidies to encourage manufacturers to invest in the production of energy-saving products and this approach has produced positive effects [7]. For example, the United States proposed an environmental quality incentive plan in the new agricultural act they enacted in 2002, which provided subsidies to promote environmental protection by producers. Within 10 years, the subsidies totaled nearly 9 billion dollars, thereby enhancing the international competitiveness of American agricultural products [8]. The State Council of China has learned from this example and in 2006 implemented interim measures for the administration of financial subsidies to promote the development of energy-efficient products, such as green air conditioners, refrigerators, and water heaters. In addition, the government has invested 26.5 billion CNY in financial subsidies, which has successfully pushed energy-saving household appliances into Chinese households [9].

These policies have greatly promoted the development of energy conservation in industry, which is not only valuable for business but also favorable for society [10]. However, different subsidy objects and ways produce different policy effects [11]. For the government, it is important to choose appropriate decision-making modes which can stimulate enterprises and consumers to produce and consume energy-saving products to the greatest extent, so as to contribute in the maximization of social welfare. At the same time, the energy-saving is related not only to industrial enterprises, but also to upstream and downstream enterprises in the process of decision-making [12]. Therefore, this paper studies the impact of government subsidy strategy on the green development of the supply chain, which refers to manufacturers and retailers jointly promoting energy conservation and emission reduction. Through the comparison of centralized decisions, decentralized decisions, and contract coordination decisions, we can help the manufacturer, retailer, and consumer to choose the optimum decision scheme.

In this context, we designed the present study to develop a new method of subsidy implementation that accounts for the needs of all stakeholders. Specifically, as follows:

(1) We simultaneously account for the product's energy-saving level, enterprise R&D costs, government subsidy costs, social welfare considerations related to the product's retail price, market demand, and profit throughout the supply chain. Accounting for these factors makes the decision-making scheme more complete.

(2) We introduced two revenue-sharing parameters to construct a contract-coordination model that achieves bidirectional sharing of profits and coordination of the supply chain. This has not been taken into account in previous studies.

(3) We established three decision models (i.e., centralized and decentralized decision-making, and the new contract-based model) and compared the optimal equilibrium solutions under the three models.

The paper is organized as follows: Section 1 introduces the writing background of the article. Section 2 presents the literature review. Section 3 describes the research questions and hypotheses and defines the problem and assumptions. Section 4 solves the models for centralized decisions, decentralized decisions, and revenue-sharing contract-coordination decisions. Section 5 provides a case study that verifies the solutions under different decision-making situations. Finally, Section 6 summarizes the results and provides recommendations for future research.

## 2. Literature Review

The green supply chain (GSC) is a modern approach to management that takes environmental impact and resource efficiency into consideration in the whole supply chain, also known as the environmental supply chain [13]. It plays an important role in the practice of green pollution prevention and green product development [14]. Game model is often used to solve the management

problems in the green supply chain [4]. Nagurney et al. [15] studied the decision-making behavior of supply chain members through game theory and established the corresponding multi-objective decision-making model. The utility function of consumers is used to construct the supply chain decision model. Liu et al. [6] used game theory to study the impact of consumers' environmental preferences on price decisions of supply chain members. It is pointed out that the improvement of consumers' environmental awareness can benefit enterprises in the green supply chain.

Although many studies have incorporated economic and environmental factors into the supply chain model, the role of the government in the implementation and development of the green supply chain is also indispensable, especially regarding to policy making [16]. In the process of supply chain management, government intervention includes subsidies, tax relief, preferential government procurement, etc. Subsidies are the most common and effective incentives [17]. Madani et al. [18] take two competitive supply chains, green and non-green, as the research object and construct a game model considering green consumption subsidy and non-green consumption tax. The paper also analyzes the impact of fiscal and taxation policies on the optimal decision in supply chains under centralized and decentralized decision-making. Bi et al. [19] study the government subsidy model to encourage enterprises to adopt green emission reduction technology and find out that the degree of green technology and the environmental improvement play important roles in the government subsidy strategy. Li and Yi [20] investigate an energy-saving incentive for products based on a benchmark energy saving set by the government and analyze the size of the government subsidy to manufacturers required to effectively balance social responsibility with corporate profit. It is concluded that energy-saving contracts can bring new opportunities to the profitability of enterprises. Lu and Shao [21] construct a joint optimal decision-making model that includes energy performance pricing, performance of government-provided production, and R&D subsidies. The researchers consider that price sensitivity and performance level sensitivity both have a significant impact on the selection of government subsidy performance level. Huang et al. [22] analyze incentives for the government to subsidize consumers to buy energy-efficient cars. Benefiting from such a scheme, each consumer can enjoy a subsidy from the government. It is shown that these incentive schemes prove to be more effective in increasing the sales of electric vehicles because they increase the bargaining power of consumers.

At the actual activity of enterprises, when they face the market risks caused by the heterogeneity of green preferences of consumers and other enterprises, supply chain members not only pursue the maximization of expected profits, but also consider the impact of profit fluctuations caused by market risks on the operations of enterprises [23]. However, the problem of energy conservation subsidies is a social problem that involves cooperation among governments, enterprises, and consumers. Only by coordinating the interests of all three stakeholders will it become possible to choose the optimal plan for all stakeholders. Therefore, contract coordination plays an important role in the supply chain of government subsidies. Ghosh and Shah [24] study a secondary supply chain composed of a manufacturer and a retailer, where market demand is jointly determined by price and green innovation and coordinated with a two-part contract. Cao et al. [25] investigate the utility and heterogeneity of green products, discuss the green supply chain coordination strategy based on the Stackelberg game and cooperative decision-making, and propose the nonlinear pricing strategy based on Nash negotiation. However, their research does not take into account the role of the government.

This literature review reveals that previous studies, which have highlighted the role of game theory and government subsidies in green supply chains, mainly focused on the methods and objectives of government subsidies, without considering the possibility of a third scenario, that is, that upstream and downstream supply chain members sign a bidirectional benefit sharing contract to achieve a mutual benefit and win-win cooperation. Therefore, when a given strategy damages the interests of one or more members of a supply chain, it's necessary to coordinate their interests from various aspects and find ways to compensate for the damage. In addition, we also need to look from the side of the government to discuss how to set the optimal proportion of government subsidies to achieve

the balance between social benefits and corporate profits and how to promote the maximization of social welfare.

## 3. Problem Definition and Assumptions

In this paper, we focus on the optimal decision-making scheme for the production and sale of energy-saving products under the premise that production will be promoted by government subsidies. For simplicity, we describe a supply chain model that consists of a single manufacturer and a single retailer. Manufacturers produce energy-efficient products and retailers sell them. The government can maximize social welfare by providing subsidies to manufacturers for energy-efficient products. The size of the market demand depends on the product price and the level of energy conservation. This problem can be solved under three different plans, as follows: Centralized (C), decentralized (D), and revenue sharing (CD) contracts. Revenue sharing (CD) combines aspects of the C and D decision-making schemes. The three schemes can be compared to reveal the most profitable one for the manufacturer and the retailer.

We defined the following main assumptions:

**Assumption 1.** *The cost for the manufacturer to produce energy-saving products is $C_m$, the seller's sales cost is $C_r$, the wholesale price is $\omega$, the retail price is p, and $p > \omega > 0$.*

**Assumption 2.** *The retail price and energy-saving level of the product are linearly related. According to our review of the literature, a typical demand function is $D(p, \lambda) = \alpha - bp + \beta\lambda$, where $\lambda$ represents the energy-saving level, $\alpha$ represents the total market demand for the energy-saving products, b represents the sensitivity of consumers to the retail price p, $\beta$ represents the sensitivity of consumers to the energy-saving level $\lambda$, and $\alpha$ and b are both greater than 0 [26–28].*

**Assumption 3.** *To improve the energy-saving level of products, manufacturers must increase their R&D investment. $G(\lambda)$ represents the R&D cost and we assume that the higher the energy-saving level the higher the R&D cost. Thus, $G'(\lambda) > 0$. We defined the R&D cost for energy-saving products using a quadratic relationship as a function of the energy-saving level of the product, $G(\lambda) = Z\lambda^2/2$, where Z represents the fixed cost invested (relatively large) in R&D for energy-saving products [4,29].*

**Assumption 4.** *The government provides production subsidies that are proportional to the energy-saving level of the products and the government for each unit of production subsidy cost is $k\lambda$, where k represents the government's subsidy intensity for the production cost of each unit of product for manufacturers and $k > 0$. This includes regulatory costs ($\eta k^2/2$), where $\eta > 0$ is the input coefficient for the government subsidy cost. This includes office costs, audits, and other input costs [30].*

**Assumption 5.** *The government provides manufacturers with subsidies to promote the production of energy-saving products, both to improve energy conservation and to maximize social welfare. The social welfare function (SW) can be expressed as the consumer surplus (CS) plus a producer surplus (PS), minus the total government subsidy cost (S) [31], as follows: SW = CS + PS − S.*

Based on these assumptions, the manufacturer's profit function ($\pi_m$) can be obtained as follows:

$$\pi_m = (\omega - c_m)(\alpha - bp + \beta\lambda) - \frac{1}{2}Z\lambda^2 + k\lambda. \tag{1}$$

The retailer's profit function is expressed as follows:

$$\pi_r = (p - \omega - c_r)(\alpha - bp + \beta\lambda). \tag{2}$$

The profit function ($\pi$) is expressed as follows:

$$\pi = \pi_m + \pi_r = (p - c_m - c_r)(\alpha - bp + \beta\lambda) - \frac{1}{2}Z\lambda^2 + k\lambda. \tag{3}$$

The functional relationship for social welfare (*SW*) is expressed as follows:

$$SW = CS - PS - S = \frac{1}{2b}D^2 + (p - c_m - c_r)D - k\lambda - \frac{\eta k^2}{2}. \tag{4}$$

## 4. Model Construction and Solutions

### 4.1. Centralized Decision Model

Centralized decision making is a theoretically ideal decision method. Its goal is to achieve the maximum profit for each member in the supply chain and thereby maximize the overall profit of the supply chain [32]. The first derivatives of Equation (3), with respect to $p$ and $\lambda$, which represent (respectively) the changes in profit per unit change of price and per unit change of the energy-saving level, are obtained as follows:

$$\partial\pi/\partial p = \alpha - bp + \beta\lambda + b(c_m + c_r - p), \tag{5}$$

$$\partial\pi/\partial\lambda = k - \lambda Z - \beta(c_m + c_r - p). \tag{6}$$

**Proposition 1.** *When* $|H| = 2bZ - \beta^2 > 0$ *is satisfied, where H represents the Hessian matrix for the supply chain profit* $\pi$ *with respect to p and* $\lambda$ *and is negative definite. The objective function has a maximum value, which is* $p^{c*}$ *and* $\lambda^{c*}$, *respectively.*

**Proof.** If $\frac{\partial^2\pi}{\partial p^2} = -2b$ and $\frac{\partial^2\pi}{\partial\lambda^2} = -Z$, it is easy to verify that the Hessian matrix of $\pi$ with respect to $p$ and $\lambda$, as follows:

$$H = \begin{bmatrix} \frac{\partial^2\pi}{\partial p^2} & \frac{\partial^2\pi}{\partial p\partial\lambda} \\ \frac{\partial^2\pi}{\partial\lambda\partial p} & \frac{\partial^2\pi}{\partial\lambda^2} \end{bmatrix} = \begin{bmatrix} -2b & \beta \\ \beta & -Z \end{bmatrix} \tag{7}$$

It can be seen from Equation (7) that the Hessian matrix is negative definite if it satisfies the condition $|H| = 2bZ - \beta^2 > 0$, which means that $\pi$ is a joint concave function of $p$ and $\lambda$ and the target function has a maximum. Thus, Proposition 1 is proved.　□

If we use a simultaneous equation model for $\frac{\partial\pi}{\partial p} = 0$, $\frac{\partial\pi}{\partial\lambda} = 0$, then the optimal retail price and energy-saving level of the energy-saving products are obtained as Equations (8) and (9), respectively, as follows:

$$p^{c*} = \frac{k\beta + \alpha Z - c_m\beta^2 - c_r\beta^2 + bc_m Z + bc_r Z}{2bZ - \beta^2}, \tag{8}$$

$$\lambda^{c*} = \frac{\alpha\beta + 2bk - \beta bc_m - \beta bc_r}{2bZ - \beta^2} \tag{9}$$

where "c" represents variables used in centralized decision-making. If we substitute Equations (8) and (9) into the market demand function $D = \alpha - bp + \beta\lambda$, the optimal market demand for energy-saving products is the following:

$$D^{c*} = \frac{b(\beta k + \alpha Z - bc_m Z - bc_r Z)}{2bZ - \beta^2}. \tag{10}$$

Equations (8), (9), and (10) can be substituted into the functional relationship expressions of Equations (3) and (4) and the total profit of the supply chain then becomes the following:

$$\pi^{c*} = \frac{(bc_m + bc_r - \alpha)^2}{4b} + \frac{(\alpha\beta + 2bk - \beta bc_m - \beta bc_r)^2}{4b(2bZ - \beta^2)}. \tag{11}$$

The optimal social welfare is:

$$SW^{c*} = \frac{\begin{array}{c}2b[\beta k + Z(\alpha - bc_m - bc_r)][\beta^2 c_r + k\beta + Z(\alpha - bc_m - bc_r)] \\ + b^3[\beta k + Z(\alpha - bc_m - bc_r)]^2\end{array}}{2(2Zb - \beta^2)^2} - \frac{k[2bk + \beta(\alpha - bc_m - bc_r)]}{2bZ - \beta^2} - \frac{\eta k^2}{2}. \tag{12}$$

### 4.2. Decentralized Decision Model

In decentralized decision making, the decision preferences of all members are risk-neutral [33]. The decision-making process is carried out according to the maximization of each stakeholder's own interests [34]. Enterprises in the supply chain form a Stackelberg game under decentralized decision-making and such a game is also a dynamic game, under the assumption of complete information [35]. In a market dominated by manufacturers, the game order as follows. First, the manufacturer determines the energy-saving level and wholesale price of the product. The retailer then determines the retail price of the energy-efficient product.

**Proposition 2.** *When* $|H| = bZ - \beta^2/4 > 0$ *such that* $4bZ > \beta^2$ *is satisfied, the Hessian matrix of the manufacturer's supply chain profit* $(\pi_m)$ *with respect to* $\omega$ *and* $\lambda$ *is negative definite. The objective function has a maximum value, which is* $\omega^{d*}$ *and* $\lambda^{d*}$*, respectively. Here, "d" represents variables used in decentralized decision-making.*

**Proof.** The solution is obtained by backward induction. First, we take the first partial derivative of Equation (2), with respect to p, to determine the change of retailer profit per unit change in the retail price, as follows:

$$\frac{\partial \pi_r}{\partial p} = \alpha + \beta\lambda - bp + b(c_r - p + \omega). \tag{13}$$

We then set equation (13) equal to zero, as follows:

$$p^d = \frac{\alpha + \beta\lambda + b(c_r + \omega)}{2b}. \tag{14}$$

If we substitute equation (14) into the manufacturer profit Function (1), we obtain the following: $\frac{\partial^2 \pi_m}{\partial \lambda^2} = -Z < 0$, $\frac{\partial^2 \pi_m}{\partial \omega^2} = -b < 0$ $\frac{\partial^2 \pi_m}{\partial \lambda^2} = -Z < 0$, $b > 0$. Thus, the Hessian matrix of $\pi$m with respect to $\omega$ and $\lambda$ is the following:

$$H = \begin{bmatrix} \frac{\partial^2 \pi_m}{\partial \lambda^2} & \frac{\partial^2 \pi_m}{\partial \lambda \partial \omega} \\ \frac{\partial^2 \pi_m}{\partial \omega \partial \lambda} & \frac{\partial^2 \pi_m}{\partial \omega^2} \end{bmatrix} = \begin{bmatrix} -Z & \frac{\beta}{2} \\ \frac{\beta}{2} & -b \end{bmatrix}. \tag{15}$$

It can be seen from Equation (15) that the Hessian matrix is negative definite. If it satisfies the condition $|H| = bZ - \beta^2/4 > 0$ such that $4bZ > \beta^2$, then the Hessian matrix is negative definite, which means that $\pi$m is a joint concave function of p and $\lambda$. The target function has a maximum and Proposition 2 is proved. The results of Propositions 1 and 2 are consistent with the research results of Jamali [4] and Modak [32]. □

For the simultaneous equations $\frac{\partial \pi_m}{\partial \lambda} = 0$ and $\frac{\partial \pi_m}{\partial \omega} = 0$, the wholesale price of energy-saving products and the energy-saving level of products under the decentralized decision-making are obtained as follows:

$$\omega^{d*} = \frac{-c_m\beta^2 + 2k\beta + 2\alpha Z + 2bc_m Z - 2bc_r Z}{4bZ - \beta^2}, \tag{16}$$

$$\lambda^{d*} = \frac{\beta(\alpha - bc_m - bc_r) + 4bk}{4bZ - \beta^2}. \tag{17}$$

By substituting the optimal wholesale price of energy-saving products and the value of the energy-saving level of the products into Equation (14), the retail price of energy-saving products becomes the following:

$$p^{d*} = \frac{-c_m\beta^2 - c_r\beta^2 + 3k\beta + 3\alpha Z + bc_m Z + bc_r Z}{4bZ - \beta^2}.$$

(18)

By substituting the retail price of the optimal energy-saving product and the energy-saving level of the product into the demand function $D = \alpha - bp - \beta\lambda$, the optimal market demand for the energy-saving product is obtained as follows:

$$D^{d*} = \frac{b(\beta k + \alpha Z - bc_m Z - bc_r Z)}{4bZ - \beta^2}.$$

(19)

By substituting the optimal values of Equations (16), (17), and (18) into Equations (1), (2), and (3), the optimal value of manufacturer's profit becomes the following:

$$\pi_m^{d*} = \frac{(\beta\alpha + 4bk - \beta bc_m - \beta bc_r)^2}{8b(4bZ - \beta^2)} + \frac{(bc_m + bc_r - \alpha)^2}{8b}.$$

(20)

The optimal profit for the retailer is the following:

$$\pi_r^{d*} = \frac{b(\beta k + \alpha Z - bc_m Z - bc_r Z)^2}{(4bZ - \beta^2)^2}.$$

(21)

The optimal value of total profit is as follows:

$$\pi^{d*} = \frac{(\beta\alpha + 4bk - \beta bc_m - \beta bc_r)^2}{8b(4bZ - \beta^2)} + \frac{(\alpha - bc_m - bc_r)^2}{8b} + \frac{b(\alpha Z + \beta k - bc_m Z - bc_r Z)^2}{(4bZ - \beta^2)^2}.$$

(22)

The optimal value of social welfare is as follows:

$$SW^{d*} = \frac{(6b + b^3)[\beta k + Z(\alpha - bc_m - bc_r)]^2}{2(4Zb - \beta^2)^2} - \frac{k[4bk + \beta(\alpha - bc_m - bc_r)]}{4bZ - \beta^2} - \frac{\eta k^2}{2}.$$

(23)

**Proposition 3.** *Compared with centralized decision-making, the retail price of energy-saving products under decentralized decision-making increases, but the market demand decreases. The energy-saving level of the products is low, the overall profit of the supply chain is reduced, and the social welfare created by government incentives and punishments is reduced.*

**Proof.** The retail price of energy-saving products in the centralized decision-making situation is subtracted from the retail price of energy-saving products in the decentralized decision-making situation to determine the price difference between these contexts, as follows:

$$p^{c*} - p^{d*} = -\frac{(2bZ - 2\beta^2)(\beta k + \alpha Z - bc_m Z - bc_r Z)}{\beta^4 + 8b^2 Z^2 - 6\beta^2 bZ}.$$

(24)

Since Z is a large number of constants, $2bZ - 2\beta^2 > 0$ and $\alpha - bC_m - bC_r > 0$, $p^{c*} - p^{d*} < 0$; that is, $p^{c*} < p^{d*}$. Thus:

$$\lambda^{c*} - \lambda^{d*} = \frac{2b\beta(\beta k + \alpha Z - bc_m Z - bc_r Z)}{\beta^4 + 8b^2 Z^2 - 6\beta^2 bZ},$$

(25)

$$D^{c*} - D^{d*} = \frac{2b^2 Z(\beta k + \alpha Z - bc_m Z - bc_r Z)}{\beta^4 + 8b^2 Z^2 - 6\beta^2 bZ}, \tag{26}$$

$$\pi^{c*} - \pi^{d*} = \frac{2b^2 Z(\beta k + \alpha Z - bc_m Z - bc_r Z)^2}{(2bZ - \beta^2)(4bZ - \beta^2)^2}, \tag{27}$$

$$SW^{c*} - SW^{d*} = \frac{2b(\beta k + \alpha Z - bc_m Z - bc_r Z)(b^2 c_m Z^2 - \beta^3 k + b^2 c_r Z^2 - \alpha bZ^2 + 3\beta bkZ)}{(2bZ - \beta^2)(4bZ - \beta^2)^2}. \tag{28}$$

Thus, $\lambda^{c*} > \lambda^{d*}$, $D^{c*} > D^{d*}$, $\pi^{c*} > \pi^{d*}$, and $SW^{c*} > SW^{d*}$. Proposition 3 is proved. □

Our analysis of Proposition 3 shows two things. First, when the total market demand is a steady state value, since the retail price of energy-saving products is the lowest and the energy-saving level is the highest under centralized decision-making, the energy-saving products will be generally welcomed by consumers and the market demand will also increase. Second, compared with centralized decision-making, decentralized decision-making has the double marginal effect of decreasing the product's energy-saving level and market demand for the product, which reduces the total profit of the supply chain.

The result of Proposition 3 verifies the research of Zhang and Wang [28]. (We all agree) It is considered that the total profit of the supply chain is higher when centralized decision is made than in the case of decentralized decisions, while the retail price of products is lower in the decentralized decision-making. The difference is in comparing the retail price of products from direct selling and retail sales. Thus, the price of products in direct selling is equal to the retail price of products in centralized decision-making. Although this paper does not consider direct sales, it includes the comparison of market demand for the product energy saving level and social welfare. Previous study does not take this into account.

From this analysis, we can infer that the result of the centralized decision-making supply chain is more nearly ideal. However, in reality, it is very difficult to achieve a completely centralized decision-making situation. Enterprises will always choose the most appropriate mode of operation according to their own operating conditions and needs and will make decisions independently. If manufacturers and retailers both make decisions in this manner, then the total supply chain profit is less than the total under centralized decision making, which results in a double marginalized effect. Moreover, under decentralized decision-making, the energy-saving level of the products and the market demand for the products are both lower. Thus, decentralized decision-making not only decreases overall profits, but it also fails to motivate manufacturers to improve the energy-saving level of their products. To solve these problems, it is obviously necessary to introduce other measures to coordinate the whole supply chain.

### 4.3. Coordination Decision Model for a Two-Way Revenue Sharing Contract

In this section, we develop a revenue-sharing contract model in which manufacturer and retailer revenues can be shared bidirectionally, with the goal of modifying decentralized decision-making to achieve the superior results of centralized decision-making through improved coordination of the supply chain. The coordination mechanism is as follows: Manufacturers and retailers enter into a contract based on the goal of ensuring profits for both parties. Manufacturers sell energy-saving products to retailers at a lower wholesale price, $\omega^{dc}$, where "dc" represents a combination of decentralized and centralized decisions and retailers transfer $(1 - \mu_1)pD$ of sales income to the manufacturer, where $\mu_1$ represents the percentage of revenue that retailers earn from selling energy-saving products after the retailer completes sales of the product. Similarly, the manufacturer transfers $(1 - \mu_2) k\lambda$ of government subsidies to the retailer, where $\mu_2$ represents the percentage of government subsidies received by

manufacturers, such that $0 < \mu_1, \mu_2 < 1$. Under the contract to ensure coordination of two-way revenue sharing, the profit functions of the manufacturer and the retailer are expressed as follows:

$$\pi_m = [\omega - c_m + (1 - u_1)(p - \omega - c_r)](\alpha - bp + \beta\lambda) - \frac{1}{2}Z\lambda^2 + (1 - u_2)k\lambda, \tag{29}$$

$$\pi_r = u_1(p - \omega - c_r)(\alpha - bp + \beta\lambda) + \mu_2 k\lambda. \tag{30}$$

Let us take the first derivative of the retailer's profit function, with respect to p, to calculate the change in revenues per unit change in price and set $\pi_r/p = 0$. The optimal retail price *p* for the energy-saving products, after contract coordination, is as follows:

$$p^{dc*} = \frac{\alpha + \beta\lambda + bc_r + b\omega}{2b}. \tag{31}$$

**Proposition 4.** *A revenue-sharing contract can effectively promote coordination of the supply chain if it satisfies the following conditions:*

$$\omega^{dc*} = C_m, \min \mu_1 < \mu_1 < \max \mu_1, \text{ and } \min \mu_2 < \mu_2 < \max \mu_2. \tag{32}$$

**Proof.** To achieve the maximum value of the optimal profit for the supply chain, while achieving profit distribution and revenue sharing through the coordination contract, $p^{dc*} = p^{c*}$, the optimal wholesale price $\omega^*dc$ for energy-saving products after contract coordination is, therefore, the following:

$$\omega^{dc*} = Cm. \tag{33}$$

If we substitute Equations (31) and (32) into Equations (29) and (30), we obtain the following equations for manufacture, retail, and total profits, respectively, as follows:

$$\pi_m^{dc*} = \frac{2k(2bZ - \beta^2)(1 - u_2)[2bk + \beta(\alpha - bc_m - bc_r)] + 2b(1 - u_1)(\beta k + \alpha Z - Zbc_m - Zbc_r)^2 - Z(\beta\alpha + 2bk - \beta bc_m - \beta bc_r)^2}{2(2bZ - \beta^2)^2}, \tag{34}$$

$$\pi_r^{dc*} = \frac{bu_1[\beta k + Z(\alpha - bc_m - bc_r)]^2}{(2bZ - \beta^2)^2} + \frac{ku_2(\beta\alpha + 2bk - \beta bc_m - \beta bc_r)}{2bZ - \beta^2}, \tag{35}$$

$$\pi^{dc*} = \frac{[2bZk - 2\beta^2k - \beta Z(\alpha - bc_m - bc_r)][2bk + \beta(\alpha - bc_m - bc_r)] + 2b(\beta k + \alpha Z - Zbc_m - Zbc_r)^2}{2(2bZ - \beta^2)^2}. \tag{36}$$

According to the contract, the coordinated contract can only be promoted when the total profit of the manufacturer and the retailer is greater than that with the decentralized decision, that is, $\pi_m^{dc*} > \pi_m^{d*}$ and $\pi_r^{dc*} > \pi_r^{d*}$.

We can then calculate the following:

$$u_1\min = \frac{(2bZ - \beta^2)^2}{(4bZ - \beta^2)^2} - \frac{ku_2(2bZ - \beta^2)(2bk + \beta A)}{b(\beta k + AZ)^2}, \tag{37}$$

$$u_1\max = \frac{(2bZ - \beta^2)(\beta^2bk^2 + A^2bZ^2 + A\beta^3ku_2 + 2bk^2u_2(\beta^2 - 4b) + 2A\beta bkZ(1 - 2u_2))}{b(4bZ - \beta^2)(\beta k + AZ)^2}, \tag{38}$$

$$u_2\min = \frac{b[2bZ(1 - 2u_1) + \beta^2(u_1 - 1)](\beta k + AZ)^2}{k(2bZ - \beta^2)(4bZ - \beta^2)(2bk + A\beta)}, \tag{39}$$

$$u_2 \max = \frac{[b(2bZ - \beta^2)^2 - bu_1(4bZ - \beta^2)^2](\beta k + AZ)^2}{k(2bk + A\beta)(2bZ - \beta^2)(4bZ - \beta^2)^2}.$$ (40)

By comparing the results of Equations (34) and (11), we conclude that $\pi^{dc*} = \pi^{c*}$. Therefore, the total profit of the green supply chain, after introduction of the contract mechanism, equals the result under centralized decision-making. Thus, Proposition 4 is proved. □

Proposition 4 shows that manufacturers can achieve supply chain coordination by selling energy-saving products at a lower wholesale price, with the decrease in revenues compensated for by the government subsidies and the proportion of the coordination between manufacturers and retailers cannot be lower than the maximum value in decentralized decision-making. Therefore, reasonable negotiation should be conducted by both parties based on the increase in their individual and shared interests.

The result of Proposition 4 proves the research of Ghosh and Shah [24]. (We all believe) It is considered that supply chain coordination can be achieved through cooperation between manufacturers and retailers and, at the same time, it can indeed improve the energy efficiency (green) level of products. Researchers claim that the improvement of the greenness can lead to the increase of the retail price of products and reduce the market demand. This paper introduces the government subsidy mechanism in the research. It means that the more energy-efficient products are, the more the government subsidizes. In this way, the loss of manufacturers is effectively compensated. Therefore, after the contract coordination between manufacturers and retailers, the retail price does not increase.

**Proposition 5.** *In the process of coordination of the supply chain, by means of a revenue-sharing contract, manufacturers and retailers can adjust the value of the contract parameters to determine the profit distribution within the supply chain. In this model, increasing the values of the contract parameters $\mu_1$ and $\mu_2$ gradually decreases the manufacturer's profit while increasing the retailer's profit proportionally.*

**Proof.** If we take the first derivatives of $\pi_m^{dc*}$ and $\pi_r^{dc*}$ to determine the profits for the manufacturer and retailer after contract coordination ($\mu_1$ and $\mu_2$, respectively), we get the following:

$$\frac{\partial \pi_m^{dc*}}{\partial u_1} = \frac{-b[\beta k + Z(\alpha - bc_m - bc_r)]^2}{(2bZ - \beta^2)^2},$$ (41)

$$\frac{\partial \pi_r^{dc*}}{\partial u_1} = \frac{b[\beta k + Z(\alpha - bc_m - bc_r)]^2}{(2bZ - \beta^2)^2}.$$ (42)

Through Equations (41) and (42), we can derive $\pi_m^{dc*}/\mu_1 > 0$ and $\pi_r^{dc*}/\mu_1 < 0$ and, similarly, $\pi_m^{dc*}/\mu_2 < 0$ and $\pi_r^{dc*}/\mu_2 > 0$. Therefore, $\pi_m^{dc*}$ is a monotonically decreasing function of the contract parameters $\mu_1$ and $\mu_2$ and the larger the values of these parameters, the smaller the value of $\pi_m^{dc*}$. We obtain the opposite results for $\pi_r^{dc*}$. Thus, Proposition 5 is proved. □

Proposition 5 adopts the research methods of Cao and Zhang [25]. It is considered that contract coordination can be achieved through the retailer revenue sharing proportion as a moderator variable. This paper adopts a two-way revenue sharing contract to coordinate the profit of the supply chain, which can increase the possibility of cooperation between both parties.

Proposition 5 shows that, for the retailer, increasing the contract parameters $\mu_1$ and $\mu_2$ decreases the sales income transferred from the manufacturer to the retailer after selling the energy-saving products, whereas the government subsidy transferred from the manufacturer to the retailer through a reduction in the wholesale price is high, thereby improving the final profit of the retailer. For the manufacturer, the wholesale price is lower than before. Therefore, increasing the distribution ratios, $\mu_1$ and $\mu_2$,

increases the impact on the wholesale price and the government subsidy income. The manufacturer's profit will also decrease with increasing values of the contract parameters $\mu_1$ and $\mu_2$.

## 5. Results and Insights

The method developed in this paper compares the equilibrium results of a green supply chain under three decision-making situations (centralized, decentralized, and coordinated control). The results let us analyze the impact of government subsidies on the retail price of energy-saving products, the market demand for the energy-saving level, and the supply chain's total profit. We can also discuss how the profits of manufacturers and retailers change and the range of values for the contract parameters before and after contract coordination. To verify the correctness of the model we have proposed, we assigned values to each parameter based on the assumptions described in the previous sections. Table 1 summarizes the resulting parameter values.

**Table 1.** Parameter values based on the discussion in Section 3.

| Parameter | $\alpha$ | $b$ | $C_m$ | $C_r$ | $Z$ | $\beta$ | $\eta$ |
|---|---|---|---|---|---|---|---|
| **Value** | 1000 | 0.8 | 30 | 3 | 50 | 0.5 | 20 |

### 5.1. The Influence of Government Subsidies on the Equilibrium Results

We analyzed the influence of government subsidies on the market demand at a given product pricing level based on the parameter values in Table 1. Figures 1–3 show the results.

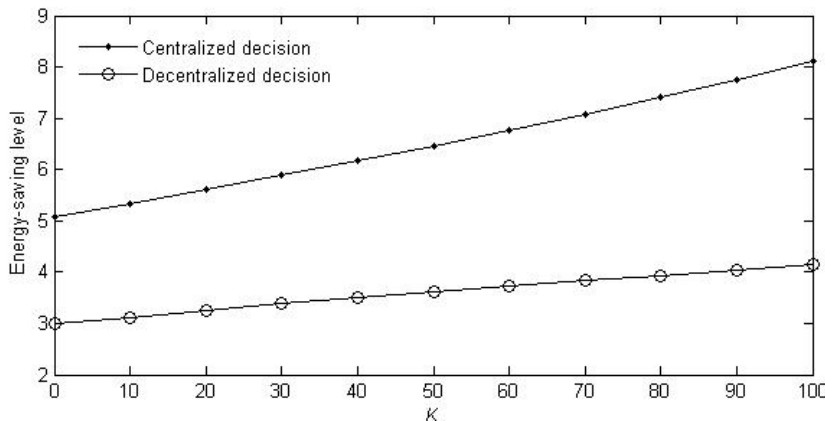

**Figure 1.** The impact of government subsidies ($k$) on the energy-saving level of products.

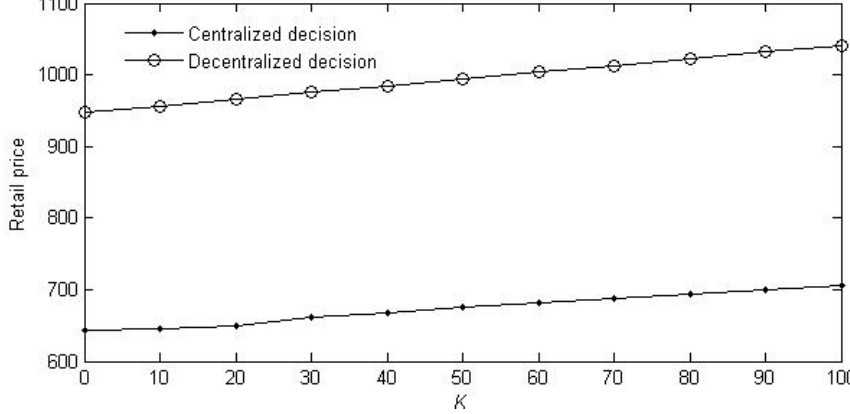

**Figure 2.** The impact of government subsidies ($k$) on the retail price of the energy-saving products.

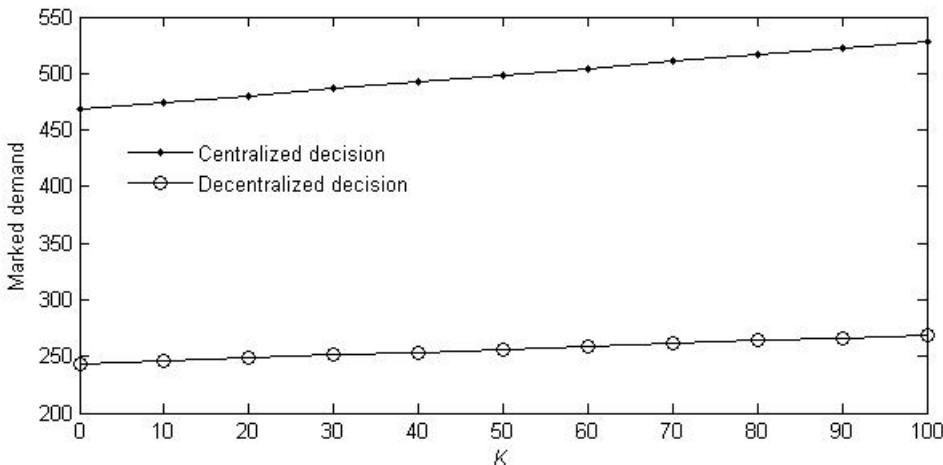

**Figure 3.** The impact of government subsidies (*k*) on the market demand for energy-saving products.

Figure 1 shows that government subsidies increase the energy-saving level of the products. The greater the government subsidy, the higher the energy-saving level. In addition, the energy-saving level is greatest under centralized decision-making.

Figure 2 shows that government subsidies have a large impact on the retail price. The retail price of energy-saving products under centralized decision-making is lower than that under decentralized decision-making and the price increases more slowly with increasing *k*.

Figure 3 shows that government subsidies have an impact on the market demand. With increasing subsidies, the market demand also increases. The market demand is greater under centralized decision-making and increases more rapidly than under decentralized decision-making.

This analysis shows that increasing government subsidies can promote energy conservation while improving demand and prices, thereby demonstrating the role of subsidies in promoting enterprise development and the production of energy-saving products. The higher the energy-saving level, the higher the retail price. At the same time, centralized decision-making lets consumers purchase goods with good quality and low prices. Therefore, the market demand will increase.

Figures 4 and 5 show the effects of government subsidies on total supply chain profits and social welfare, respectively, based on the values in Table 1.

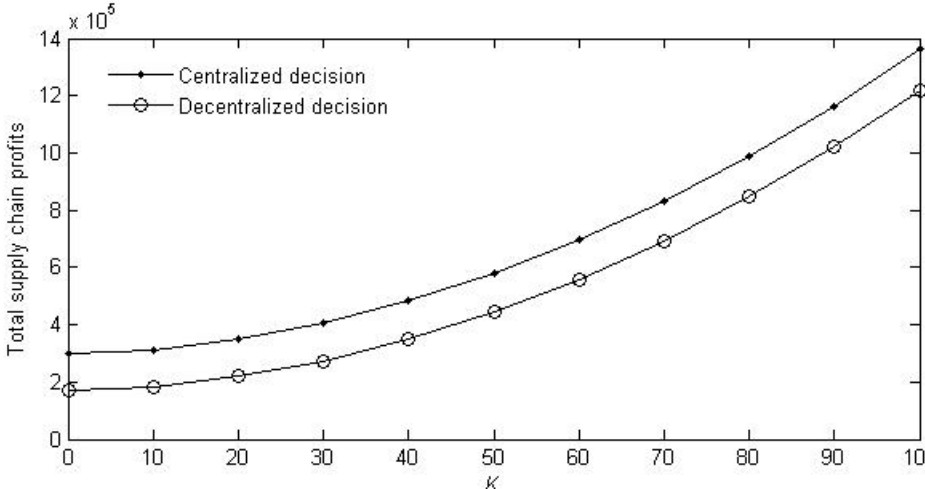

**Figure 4.** The impact of government subsidies (*k*) on total supply chain profits for energy-saving products.

Figure 4 shows that government subsidies have a significant impact on the supply chain profit. The larger the government subsidy, the higher the total supply chain profit. The total profit is also higher under centralized decision-making, but increases at the same rate in both forms of decision-making. In addition, manufacturers and retailers can increase the total profit of the supply chain in decentralized decision-making by means of contracts that coordinate profit sharing until the result is the same as that achieved by centralized decision-making.

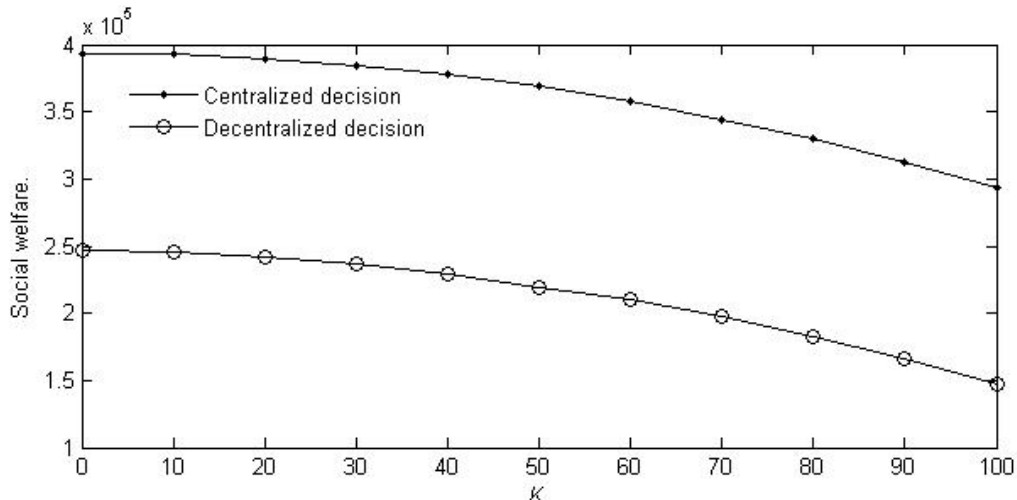

**Figure 5.** The impact of government subsidies (*k*) on social welfare for energy-saving products.

Figure 5 shows that government subsidies have a significant impact on social welfare. With increasing subsidies, social welfare will decrease and the decrease will be faster under decentralized decision-making. The social welfare created by government subsidies is higher under centralized decision-making. Figures 1–5 provide verification that Proposition 3 is valid.

### 5.2. The Effect of a Revenue Sharing Contract on the Equilibrium Results

Figures 6 and 7 show the influence of the contract parameters $\mu_1$ and $\mu_2$ on the equilibrium results for the manufacturer and retailer profits.

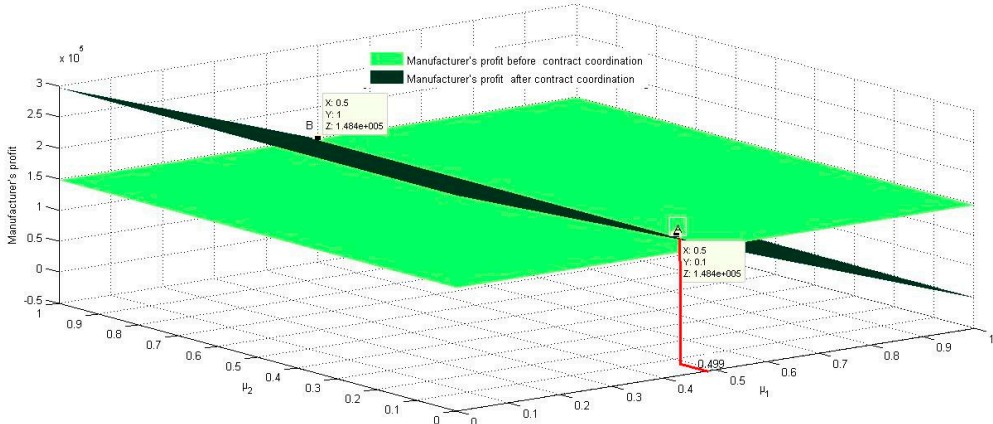

**Figure 6.** Manufacturer's profit before and after contract coordination.

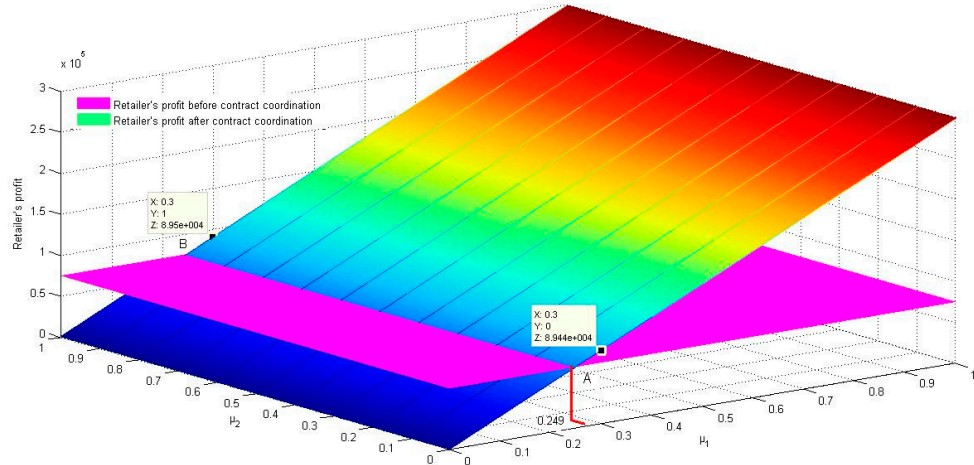

**Figure 7.** Retailer's profit before and after contract coordination.

The red lines that show the position of points A and B indicate the line formed where the two response surfaces for profit intersect. Figure 6 shows that the manufacturer's profit after contract coordination decreases with increasing $\mu_1$, but does not change in response to changes in $\mu_2$. When $\mu_1 < 0.499$, the manufacturer's profit, after contract coordination, is greater than that before contract coordination.

The red lines that show the position of points A and B indicate the line formed where the two response surfaces for profit intersect. Figure 7 shows that, after the coordination contract for revenue-sharing, the retailer's profit increases with increasing $\mu_1$ until it achieves the same result as centralized decision-making, but does not change with changes in $\mu_2$. When $\mu_1 > 0.249$, the profit for the retailer in the coordinated method is greater than that of the retailer before coordination. Figures 6 and 7 together provide verification that Proposition 5 is correct.

Our analysis in Figures 6 and 7 suggests that the profit-sharing ratio for retailers should be $0.249 < \mu_1 < 0.499$. According to Equations (39) and (40), the value range for $\mu_2$ can be calculated as $0.00417 < \mu_2 < 1$. Proposition 4 is therefore verified.

Table 2 compares the game's equilibrium results under the three following decision-making modes: Centralized, decentralized, and decentralized with a coordination contract. Let's pick the middle of the range between $\mu_1$ and $\mu_2$, for parameter values of $\mu_1 = 0.374$ and $\mu_2 = 0.502$.

**Table 2.** Equilibrium game results under the three decision-making modes.

| Variable | Centralized Model | Decentralized Model | Coordination Contract Model |
|----------|-------------------|---------------------|-----------------------------|
| $\omega$ | — | 639.5149 | 30 |
| $p$ | 643.3611 | 947.2723 | 643.3611 |
| $\lambda$ | 6.3047 | 3.2476 | 6.30470 |
| $q$ | 488.3762 | 243.8059 | 488.3762 |
| $\pi_r$ | — | 74,301.67 | 111,765.7 |
| $\pi_m$ | — | 148,372.2 | 185,442.8 |
| $\pi$ | 297,208.5 | 222,673.9 | 297,208.5 |
| $SW$ | 296,145.4 | 221,641.4 | 296,145.4 |

Table 2 provides 3 main findings. First, the sales price of energy-saving products increased by 47.3% under decentralized decision-making, the market demand for the product decreased by 50.1%, and the total supply chain profit decreased by 25.1%. Thus, the decentralized decision-making supply chain appears to exhibit a double marginal effect and the results agree with Proposition 3. Second, the profits of the manufacturer increased by 37,070.6 and the retailer's profits increased by 37464.03 as a result of coordination of the supply chain by means of the two-way revenue-sharing contract model.

This result is consistent with the line that connects points A and B in Figures 5 and 6. Third, the total profit and social welfare increased by 33.5% and 33.6%, respectively. Thus, the revenue-sharing contract can improve the performance of the supply chain considerably, and Proposition 4 is also verified.

*5.3. Implications for Theory and Management*

Through the two-way sharing of profits and government subsidies, retailer enthusiasm to sell energy-saving products can be increased, leading to increased sales volume. At the same time, if manufacturers involve retailers in the R&D phase, products can be made more attractive to potential purchasers, thereby increasing both demand and profits for both stakeholders. Although retailers may have to bear additional production costs, increased sales should lower the wholesale price of the products and maintain or improve their profits. Manufacturers and retailers can both seize new market opportunities and increase their competitiveness. For manufacturers, including input from retailers in the development of energy-saving products should not risk a decrease in the manufacturer's profits, since the R&D costs are shared and sales should increase enough to cover any additional costs. Manufacturers and retailers should work together to define an appropriate profit-sharing mechanism (i.e., a contract). This mechanism should be based on the ability of manufacturers and retailers to design and sell products that are sufficiently attractive to consumers to achieve a win–win situation. Therefore, this innovative new perspective applies not only to retail, but also to industry or services [36].

## 6. Conclusions and Suggestions for Future Research

In this study, we found a way to improve profits in a green supply chain when the government provides production subsidies for manufacturers and the subsidies can be shared with retailers through a reduction in the wholesale price. To do so, we developed decision models based on centralized, decentralized, and contract-based two-way revenue sharing. By solving for each variable, we were able to compare the equilibrium results for these models and test the influence of government subsidies and coordination contracts on the equilibrium results, using sample data to provide an example of how manufacturers and retailers can work together to improve their decisions on the production and sales of energy-saving products.

Under decentralized decision-making, the double marginal effect of the energy-saving level and market demand decrease simultaneously, thereby decreasing total profits for the whole supply chain. To improve the total profits and avoid the difficulties created by centralized decision-making in real life, we developed a coordinated decision model for two-way profit sharing between manufacturers and retailers. This novel model has not been proposed in previous studies of supply chain decision-making processes. Our analysis revealed the following: (1) The energy-saving level, market demand, and total supply chain profit for energy-saving products were highest under centralized decision-making. However, the retail price of energy-saving products was highest in decentralized decision-making. (2) With increasing government subsidies, the market demand, the energy-saving level of retail products, and the total profit of supply chain increase. (3) An income-sharing contract between manufacturers and retailers improved the total supply chain profit so that the profit under decentralized decision-making increased to the same level as in centralized decision-making. (4) Thus, to encourage consumers to purchase more energy-saving products, thereby improving profits for both manufacturers and retailers, both stakeholders should implement revenue sharing. The revenue sharing coefficient $\mu_1$ ranged between 0.249 and 0.499. The revenue sharing coefficient $\mu_2$ ranged between 0.00417 and 1.

Future research should address the following aspects: (1) Our research did not consider a situation in which multiple manufacturers and retailers compete, so future research should consider a competitive game rather than a cooperative game and determine how to set the model parameters to minimize government subsidies and maximize social welfare. (2) Our research did not consider the behavioral characteristics of consumers. The next step in this research could be to conduct a more in-depth analysis that accounts for consumer behaviors, such as marginal utilities, so as to make the model more consistent with the real-world decision-making environment. (3) It would also be

interesting to consider the possibility of subsidies to encourage recycling older and less-energy-efficient products so that consumers could replace them with energy-saving products, thereby extending our analysis in the direction of life-cycle analysis by accounting for the whole process of production, circulation, sales, recycling, and remanufacturing of energy-saving products (i.e., the whole supply chain). (4) Our research does not take into account the factors of changes in the market economy. In the future, we will focus on the impact of government subsidies on market economy, especially on the market price of products. We should further analyze the changes of market managers and policy makers in management and policy formulation brought by subsidy policies.

**Author Contributions:** This paper was written by R.G. in collaboration with all co-authors. The formal was analyzed by L.Z. The funding was acquired by J.X. Data was calculated by X.J. The formal was analyzed by Y.X.

**Funding:** This study was supported by grants from the National Social Science Foundation of China (Project No. 16BGL146), (Project No. 18AZD005, the National Natural Science Foundation of China (Project No. 71874108), and the Natural Science Basic Research Plan in Shaanxi Province of China (Project No. 2018JM7005).

**Conflicts of Interest:** The authors declare no conflicts of interest.

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
