# Peer review of "A Green Supply-Chain Decision Model for Energy-Saving Products That Accounts for Government Subsidies"

_sustainability, doi:10.3390/su11082209_

Round 1

Reviewer 1 Report

I am glad to review this paper, thanks the editor for the opportunity.

The paper is related to interesting topic but it has many areas to improve:

-          There is no literature review part, the authors try to relate to literature sources in the introduction but it is very narrow. I foresee to prepare serious literature review on the studied problem in other countries

-          The proposed model and assumptions are not validated with the real circumstances. Now the results are “art for art”. The implications are so simplified that there was no need to prepare so sophisticated experiment to write the results.

-          Results are not even compared to any previous solutions from the literature

-          The authors should also consider how the subsidies influence free market economy as well

Author Response

The paper is related to interesting topic but it has many areas to improve.

Response: We appreciate your recognition of our manuscript, which is a great encouragement to us. We also would like to thank you for your valuable and constructive comments, which are of great help in refining our manuscript.

Point 1:   There is no literature review part, the authors try to relate to literature sources in the introduction but it is very narrow. I foresee to prepare serious literature review on the studied problem in other countries.

Response 1: According to your suggestion, we have supplemented a separate literature review part (see Part 2, lines 94-153 of the revised manuscript). For the problem that you proposed the scope of the literature is narrow, we’ve broadened the literature sources from the following four aspects.

 (1) Firstly, this article introduces the definition of green supply chain. On this basis, we propose a game theory approach which is often used to solve the problem of green supply chain management, and list some classic literatures in this field (see lines 94-106 of the revised manuscript).

(2) Secondly, we integrated the literature on government subsidies for the production and consumption of energy-saving products. This paper focuses on the analysis of the supply chain management in the process of government subsidies for energy-saving products to the research purposes, research methods and the effect of subsidies. (see lines 107-129 of the revised manuscript).

(3) Thirdly, combining the above two parts, we focus on the benefit coordination of supply chain management in the application of game theory. (see lines 130-143 of the revised manuscript).

(4) Finally, we summarize the research contents of the above three aspects. Through the comparison of the literature, we find out the specific problems to be solved in this paper. This paper focuses on the study of green supply chain management on the basis of government subsidies. With the method of game theory, this paper constructs a game model which can realize bidirectional coordination of supply chain profit from various aspects and levels. By comparing and analyzing the models of centralization and decentralization and contract coordination, we can find out the optimal decision scheme. (see lines 144-153 of the revised manuscript).

Point 2:  The proposed model and assumptions are not validated with the real circumstances. Now the results are “art for art”.

Response 2: For what you have put forward, “The proposed model and assumptions are not validated with the real circumstances. Now the results are “art for art””, we will make a detailed explanation.

(1) Firstly, our research hypothesis are based on references to relevant literature and sufficient literature support in this field (see lines 166-188 of the manuscript). Therefore, we have not tested it again.

(2) Secondly, we also want to test the results in the real world, but unfortunately the government departments did not make the data public. Therefore, we can only choose the mode of assignment to simulate the model results to verify the validity of the model.

(3) Thirdly, We found some literatures using similar research methods with ours. These literatures also used data allocation like ours to test the solution results of the model without verifying the effectiveness of the model in the real environment.

For example, (Pal et al., 2018 ; Han et al., 2018; Jamali et al., 2018; Chen et al., 2019) .

(i) Pal, B.; Sana, S. S.; Chaudhuri, K. Supply chain coordination with random yield and demand uncertainty. International Journal of Management Science and Engineering Management. 2018, 13 (01), 33-44.

(ii) Han, Q.,; Wang, Y.Y. Decision and Coordination in a Low-Carbon E-Supply Chain Considering the Manufacturer’s Carbon Emission Reduction Behavior. Sustainability. 2018, 10 (05), 1686-1694. 

(iii) Jamali, B.M.; Barzoki, M.R. A game theoretic approach for green and non-green product pricing in chain-to-chain competitive sustainable and regular dual-channel supply chains. Journal of Cleaner Production. 2018, 170, 1029-1043.

(iiii) Chen, Z.S.; Su, S. Social welfare maximization with the least subsidy: Photovoltaic supply chain equilibrium and coordination with fairness concern. Renewable Energy. 2019, 132, 1332-1347.

(4) According to the research hypothesis. We set the parameter values in the manuscript. We hope to illustrate the feasibility of the solution and make the results of the study more intuitive by a numerical example.

Point 3:  The implications are so simplified that there was no need to prepare so sophisticated experiment to write the results.

Response 2: For what you have put forward, “The implications are so simplified that there was no need to prepare so sophisticated experiment to write the results.”, we also will make a detailed explanation.

(1) Firstly, in this paper, we hope to choose the optimal decision-making scheme through the comparison of various schemes. Therefore, we set up three decision-making schemes. Through the comparison of the results, it is found out that among the centralized decision-making schemes, the retail price of the product is the lowest, and the energy saving level is the highest, supply chain profits and social welfare are higher than decentralized decision-making. However, the centralized decision-making scheme is an ideal one. In reality, it is difficult to implement as long as there is a member of the supply chain with different opinions. Decentralized decision-making scheme is faced with the problem of high product price, low market demand and low supply chain profit. Therefore, it is necessary to adopt a new scheme to coordinate the profit of the supply chain, in order for it to achieve the effect of centralized decision-making under the decentralized decision-making situation, improve the market demand of products, and protect the interests of the government, enterprises and consumers from being affected. So the experiment is going to look complicated.

(2) Secondly, the following articles do the same:

(i) Aslani, A.; Jafar, H. Transshipment contract for coordination of a green dual-channel supply chain under channel disruption. Journal of Cleaner Production. 2019, 223, 596-609.

(ii) Savaskan, R. C.; Wassenhove, B. L. Closed-loop supply chain models with product remanufacturing. Management science. 2004, 50 (2), 239-252.

(iii) Modak, N. M; Panda , S.; Sana, S. S. Pricing policy and coordination for a two-layer supply chain of duopolistic retailers and socially responsible manufacturer. International Journal of Logistics Research and Applications, 2016, 19 (6), 487-508.

Point 4: Results are not even compared to any previous solutions from the literature.

Response 4: For what you have put forward, “Results are not even compared to any previous solutions from the literature”. It has been modified according to your opinion.

We compare all the results of propositions 1-5 with the previous results, and explain the differences between our results and the previous studies, respectively (see lines : 254-256;302-309; 371-379; 393-396).

The following content is modified:

(1) The results of propositions 1 and 2 are consistent with the research results of Jamali [4] and Modak [35]. Only when the Hessian matrix is negative, the result of the supply chain function relation can have the maximum value.

(2) The result of proposition 3 verifies the research of Zhang and Wang [30]. (We all agree) It is considered that the total profit of the supply chain is higher when centralized decision is made than in the case of decentralized decision, while the retail price of products is lower in the decentralized decision-making. The difference is in comparing the retail price of products from direct selling and retail sales. Thus, the price of products in direct selling is equal to the retail price of products in centralized decision-making. Although this paper does not consider direct sales, it includes the comparison of market demand for product energy saving level and social welfare. Previous study does not take it into account.

(3) The result of proposition 4 proves the research of Ghosh and Shah [25]. (We all believe) It is considered that supply chain coordination can be achieved through cooperation between manufacturers and retailers, and at the same time, it can indeed improve the energy efficiency (green) level of products. Researchers claim that the improvement of the greenness can lead to the increase of the retail price of products and reduce the market demand. This paper introduces the government subsidy mechanism in the research. It means that the more energy-efficient of manufacturer products, the more government subsidies. In this way, the loss of manufacturers is effectively compensated. Therefore, after the contract coordination between manufacturers and retailers, the retail price does not increase.

(4) Proposition 5 adopts the research methods of Cao and Zhang [27]. It is considered that contract coordination can be achieved through the retailer revenue sharing proportion as a moderator variable. this paper adopts two-way revenue sharing contract to coordinate the profit of supply chain, which can increase the possibility of cooperation between both parties.

 Point 5: The authors should also consider how the subsidies influence free market economy as well.

Response 5: For what you have put forward, “The authors should also consider how the subsidies influence free market economy as well”. It has been modified according to your opinion.

You provide a very good perspective. Indeed, as you said, the way of government subsidies will have an impact on the market economy to some extent. So how does it work? Can extent of the impact be measured? These questions are all very interesting and worth studying. We can write a new manuscript according to your ideas. In the future, we will focus on the impact of government subsidies on the market economy, especially on the market price of products, although we can't consider all of them in our manuscript.

The following content is modified

(4) Our research does not take into account the factors of changes in the market economy. In the future, we will focus on the impact of government subsidies on market economy, especially on the market price of products. We should further analyze the changes of market managers and policy makers in management and policy formulation brought by subsidy policies.

Reviewer 2 Report

I was pleased to read your study and I believe it raises interesting insights. The literature section brings robust arguments based on references that support the originality of the study. The paper deals with a topic of key interest and the content of the paper seems adequate for the purposes of the journal. The authors clearly explain the details of centralized and decentralized decision-making models for the green supply chain and a coordinated decision-making model for revenue-sharing contracts based on game theory. Nevertheless, some revisions are required and I think that authors need to carefully consider the points detailed below to improve the quality of the paper. My main suggestions are the following. The introduction should underline the aim of the paper, stress why this aim is important (with the support of the literature), summarise the methodology, and how the paper is organised. The authors should stress the aim of the research and discuss the importance of this topic. As for the introduction and the background of the study, I suggest to improve this section starting to analyse the recent contributions on the topic of environmental sustainability orientation (Croom et al., 2018; Jin et al., 2018; Shashi et al., 2018) and then focus on the topic of sustainability orientation in the context of global supply chains. Croom et al. (2018). Impact of social sustainability orientation and supply chain practices on operational performance. International Journal of Operations and Production Management, 38 (12), pp. 2344-2366. Shashi et al. (2018) Sustainability orientation, supply chain integration, and SMEs performance: a causal analysis (2018) Benchmarking, 25 (9), pp. 3679-3701. Jin, Z., Navare, J., Lynch, R. The relationship between innovation culture and innovation outcomes: exploring the effects of sustainability orientation and firm size. R and D Management. Finally, with regard to the implications, despite the results of the proposed paper are in-depth discussed, some additional implications for future researchers, managers, policy makers should be included. Some additional limitations can be included.

Author Response

I was pleased to read your study and I believe it raises interesting insights. The literature section brings robust arguments based on references that support the originality of the study. The paper deals with a topic of key interest and the content of the paper seems adequate for the purposes of the journal. The authors clearly explain the details of centralized and decentralized decision-making models for the green supply chain and a coordinated decision-making model for revenue-sharing contracts based on game theory.

     Response: We appreciate your recognition of our manuscript, which is a great encouragement to us. We also would like to thank you for your valuable and constructive comments, which are of great help in refining our manuscript.

Point 1: The introduction should underline the aim of the paper, stress why this aim is important (with the support of the literature), summarise the methodology, and how the paper is organised. The authors should stress the aim of the research and discuss the importance of this topic.

Response1: Thank you for your valuable advice. According to your Suggestions, we have emphasized the research purpose in the introduction, and the first sentence clearly puts forward that the issue we study is the issue of sustainable development. The references are from the second paper you provided (Shashi et al., 2018) (See lines 34-35 of the manuscript).

To further illustrate the importance of the problem and the methodology, we have adopted the following approach:

(1) We illustrate the importance of this topic by listing two cases (see lines 53-62). The way of government subsidies is not only conducive to the development of enterprises but also can achieve great social benefits. A further introduction of the purpose of this manuscript concerns studying. It is to study the influence of government subsidy strategy on the green development of supply chain ,and realize the goal of energy conservation and emission reduction through the joint efforts of government enterprises and consumers.

(2) For the method, this paper uses the game model to construct three kinds of decision plans. The optimal decision scheme is selected by comparing centralized decision, decentralized decision and contract coordination decision.

(3) In order to clearly illustrate the importance of the research objectives and methods in this paper, we rewrote the literature review and listed it separately as the second part of the manuscript (see lines 94-153 of this manuscript). The application of green supply chain and game theory, the purpose and effect of government subsidy and supply chain contract coordination are analyzed in detail. And through the literature comparison, research questions and design of research scheme can be worked out. (see lines 144-153 of this manuscript) and the design of research scheme (see lines 75-86 of this manuscript).

The following content is modified:

These policies have greatly promoted the development of the energy conservation in industry, which is not only valuable for business but also favorable for society [10]. However, different subsidy objects and ways produce different policy effects [11]. For the government, it is important to choose appropriate decision-making mode which can stimulate enterprises and consumers to produce and consume energy-saving products to the greatest extent so as to contribute in the maximization of social welfare. At the same time, the energy-saving is related not only to industrial enterprises, but also to upstream and downstream enterprises in the process of decision-making [12]. Therefore, this paper studies the impact of government subsidy strategy on the green development of supply chain which refers to manufacturers and retailers jointly promoting energy conservation and emission reduction. Through the comparison of centralized decision, decentralized decision and contract coordination decision, we can help the manufacturer, retailer and consumer to choose the optimum decision scheme.

The paper is organized as follows: Section 1 introduces the writing background of the manuscript. Section 2 presents the  literature review . Section 3 describes the research questions and hypotheses, and defines the problem and assumptions. Section 4 solves the models for centralized decisions, decentralized decisions, and revenue-sharing contract-coordination decisions. Section 5 provides a case study that verifies the solutions under different decision-making situations. Finally, Section 6 summarizes the results and provides recommendations for future research.

Point 2: About the three documents you gave and your suggestions for writing.

Response 2: Thank you for your valuable comments on our manuscript. As you pointed out “your suggestions for writing”. The questions in this section, as well as the references and suggestions you have provided, are very important to improve this manuscript.

We have supplemented and improved the manuscript according to your Suggestions. The impact of government subsidies on a free market economy is added to the research and outlook. At the same time, the influence on policy makers and managers should be considered in future studies (See lines 543-546 of this manuscript). In addition, we have carefully studied the three papers you provided and quoted them into our manuscript.(Shashi et al., 2018, See lines 34-35;Croom et al., 2018 , 63-64;Jin et al., 2018; 506-507).

The following content is modified:

(1) From the perspective of theory and practice, sustainable development is a very important issue. The concept and problems of sustainability are realized in many supply chain relationships [1].

(2) These policies have greatly promoted the development of the energy conservation in industry, this is not only valuable for business but also favorable for society [10]. However, different subsidy objects and ways produce different policy effects [11].

(3) Therefore, this innovative new perspective applies not only to retail, but also to industry or services [38].

       (4) Our research does not take into account the factors of changes in the market economy. In the future, we will focus on the impact of government subsidies on market economy, especially on the market price of products. We should further analyze the changes of market managers and policy makers in management and policy formulation brought by subsidy policies.

Reviewer 3 Report

The paper “A green supply-chain decision model for energy-saving products that accounts for government subsidies” shows that government subsidies can significantly improve social welfare and promote the improvement of energy-saving products. The article seems to be suitable for the purposes of the journal.

Title: The title of the paper is informative. It includes important terms and the message of the article.

Abstract: The abstract describes the context and follow the structure: backgrounds, methods, results and conclusions.

Keywords: They are well chosen.

Introduction and literature review. Introduction defines the focus of the article. The literature review supports to understand the correlation of presented research results with literature. A summary table comparing the contributions could support the explanation (see Fig 8 in https://doi.org/10.3390/ijerph16040634)

Materials and methods. In my opinion, the methodology and the research results are extensively discussed. The key functionality has been explained but the computer application for implementation is not revealed. The details of software engineering part (pseudocode) would be interesting for anyone aiming to replicate the implementation if suitable. Scenarios validate the model.

Discussion and conclusions: Clear and adequate.

Please increase the resolution of Figures, where possible (especially Fig 6 and 7).

Author Response

The paper “A green supply-chain decision model for energy-saving products that accounts for government subsidies” shows that government subsidies can significantly improve social welfare and promote the improvement of energy-saving products. The article seems to be suitable for the purposes of the journal.

Response: we appreciate your recognition of our manuscript, which is a great encouragement to us. We also would like to thank you for your valuable and constructive comments, which are of great help in refining our manuscript.

Point 1: Moderate English changes required

Response: we invited professionals to modify the language.

Point 2Materials and methods. In my opinion, the methodology and the research results are extensively discussed. The key functionality has been explained but the computer application for implementation is not revealed. The details of software engineering part (pseudocode) would be interesting for anyone aiming to replicate the implementation if suitable. Scenarios validate the model.

Response : Thank you for your valuable advice. According to your suggestion, I uploaded all the program codes of the computer, and all the results can be copied. See attachment file for details: figure and pseudocode.

Point 3: Please increase the resolution of Figures, where possible (especially Fig 6 and 7).

Response :I have readjusted all the images and I have provided you with an image that has a minimum resolution of 300 dots per inch (dpi) and a proper print size. I have modified  Fig 6 and 7 ( in lines 459 and 467 of the revised manuscript) and uploaded the original picture of figure 1-7, please refer to the document for details: figure and pseudocode.

Round 2

Reviewer 1 Report

Dear Editor,

I am glad of the improvements and authors explanation especially. Their explanation allow me to understand their aproach better. 

I think now it can be published.

Reviewer 3 Report

Authors have made all required changes. The article can be published in this revised form.